# Visuo-motor updating in individuals with heightened autistic traits

Antonella Pomè*, Eckart Zimmermann

Institute for Experimental Psychology, Heinrich Heine University Düsseldorf, Düsseldorf, Germany

**Abstract** Autism spectrum disorder (ASD) presents a range of challenges, including heightened sensory sensitivities. Here, we examine the idea that sensory overload in ASD may be linked to issues with efference copy mechanisms, which predict the sensory outcomes of self-generated actions, such as eye movements. Efference copies play a vital role in maintaining visual and motor stability. Disrupted efference copies hinder precise predictions, leading to increased reliance on actual feedback and potential distortions in perceptions across eye movements. In our first experiment, we tested how well healthy individuals with varying levels of autistic traits updated their mental map after making eye movements. We found that those with more autistic traits had difficulty using information from their eye movements to update the spatial representation of their mental map, resulting in significant errors in object localization. In the second experiment, we looked at how participants perceived an object displacement after making eye movements. Using a transsaccadic spatial updating task, we found that those with higher autism scores exhibited a greater bias, indicating under-compensation of eye movements and a failure to maintain spatial stability during saccades. Overall, our study underscores efference copy's vital role in visuo-motor stability, aligning with Bayesian theories of autism, potentially informing interventions for improved action–perception integration in autism.

**\*For correspondence:**
antonella.pom@gmail.com

**Competing interest:** The authors declare that no competing interests exist.

## eLife assessment

This **important** study shows that a high autism quotient in neurotypical adults is associated with suboptimal motor planning and visual updating after eye movements, suggesting a disrupted efference copy mechanism. The implication is that abnormal visuomotor updating may contribute to sensory overload - a key symptom in autism spectrum disorder. The evidence presented is **convincing**, with few limitations, and should be of broad interest to neuroscientists at large.

## Introduction

Autism spectrum disorders (ASDs) present a complex array of challenges, with sensory sensitivities, a recent addition to diagnostic criteria (*Diagnostic and statistical manual of mental disorders: DSM-5, 2013*), standing out as a dominant aspect of this condition. Autistic people often exhibit a general inflexibility in the motor domain and struggle to learn from motor errors, usually linked to restricted and repetitive behaviors (*Lopez et al., 2005*; *Van Eylen et al., 2011*; *Pomè et al., 2023*). This difficulty in understanding motor errors can hinder their ability to adapt to new sensory information, further exacerbating sensory overload (*Pomè et al., 2023*; *Johnson et al., 2013*; *Mosconi et al., 2013*).

Recent research has increasingly emphasized the role of predictive abilities in ASD (*Sinha et al., 2014*; *van Boxtel and Lu, 2013*; *Van de Cruys et al., 2014*; *Pellicano and Burr, 2012*). In their daily lives within an unpredictable and often chaotic world, humans are consistently engaged in anticipating

what might happen next and preparing adaptively for potential threats in their surroundings (*Diagnostic and statistical manual of mental disorders: DSM-5, 2013*). Within this framework, recent theories propose that individuals with ASD may demonstrate a reduced reliance on prior information when making perceptual judgments. The Bayesian perspective suggests that the predictive system deficits in autism stem from a diminished Bayesian prior (*Pellicano and Burr, 2012*; *Angeletos Chrysaitis and Seriès, 2023*). From this perspective, the brain unconsciously infers information about the world by applying Bayes' rule: it assesses the probability of a prediction given sensory data, considering the likelihood of sensory data given that prediction and the Bayesian prior. The Bayesian prior is essentially a probabilistic distribution representing the expectation of the environment being in a particular state before any observations are made (*Pellicano and Burr, 2012*). These predictive abilities include the role of perceptual priors, which play a crucial role in guiding individuals to make accurate predictions, especially when sensory data is limited. However, in cases of autism, the influence of prior knowledge is believed to be weakened, resulting in predictions that heavily rely on sensory input.

But how do these priors interact with the sensory and motor systems in autistic people, and how does this influence their perceptual stability? Predicting the consequences of one's behavior is of vital importance when our movements produce changes on our sensory receptors. When we move our eyes, head, or body, we displace external stimuli on the retina. We must know about this self-produced displacement to avoid confusion with external stimulus motion. In other words, if we would not be able to predict the sensory consequences of our behavior, we would experience that the world is moving. The most frequent movements we perform are voluntary saccades. To rapidly direct the fovea, the area of highest resolution, toward regions of interest, we shift the eyeball with high speed about 3 times per second. Internal predictions about an imminent saccade ensure the smooth and stable continuity of our perception. This anticipation involves a signal incorporating information about the motor vector to predict the sensory consequences of the upcoming saccade. Such a signal is known as an efference copy, that is a copy of the motor command (for a review see *Wurtz, 2018*). During a tennis game, rapid oculomotor saccades are employed to track the high-velocity ball across the visual display. In the absence of a functional efference copy mechanism, the brain would encounter difficulty in anticipating the precise retinal location of the ball following each saccade. This could result in a transient period of visual disruption as the visual system adjusts to the new eye position. The efference copy, by predicting the forthcoming sensory consequences of the saccade,

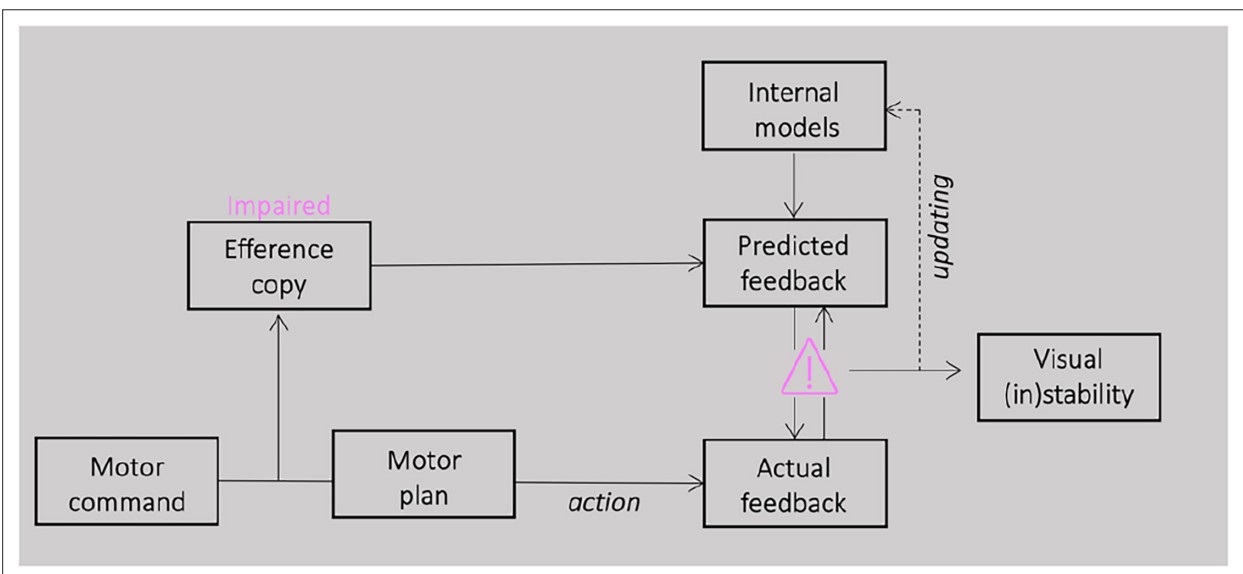

**Figure 1.** Schematic illustration of the proposed model. For saccade generation, a motor command is sent to the motor plan which produces the desired eye movement. A correct plan of the movement depends on a comparison between the position of the eye before and after a saccade and therefore on the accuracy of the efference copy signal. If the efference copy is impaired, pre-saccadic prediction will be never completely confirmed by the post-saccadic feedback. This constant lack of successful match between prediction and sensory evidence could influence the accumulation of sensory predictions over time, leading to a less stable perception.

would bridge this gap and facilitate the maintenance of a continuous and accurate representation of the ball's trajectory. A key pathway responsible for transmitting efference copy signals involves the superior colliculus (*Sommer and Wurtz, 2002*), a saccade command area, to the frontal eye field via the medio-dorsal nucleus of the thalamus. Lesions affecting this pathway can impair both motor and visual updating (*Wurtz, 2018*). We recently reported that in patients with lesions in thalamic nuclei, evidence for two pathways conveying efference copies exists. We found that patients with lesions in the medio-dorsal nucleus showed impairments in visual updating, while those with lesions in both the medio-dorsal and ventrolateral (VL) nuclei exhibited issues with motor updating (*Bellebaum et al., 2005a*; *Bellebaum et al., 2005b*; *Bellebaum et al., 2006*; *Ostendorf et al., 2013*; *Ostendorf et al., 2010*; *Zimmermann et al., 2020*).

Here, we hypothesized that the impairment of efference copy signals would result in an enduring lack of complete confirmation for pre-saccadic predictions through post-saccadic feedback. A failure in processing efference copy information might have consequences for perception around the execution of saccades. Their reliance on post-saccadic sensory feedback, rather than accurate efference copy predictions, results in a distorted perception of the visual world following eye movements (see *Figure 1*).

First, intra-saccadic vision might be affected. We never perceive the self-produced motion stimulation on the retina produced by our own saccades. Some theories consider saccadic omission and saccadic suppression as resulting from an active mechanism. In this view, an efference copy would signal the occurrence of a saccade, yielding a transient decrease in visual sensitivity (*Burr et al., 1994*; *Binda and Morrone, 2018*; *Zimmermann and Lange, 2022*). Others however have pointed out the possibility that a purely passive mechanism suffices to induce saccadic omission (*Castet et al., 2002*). A recent study has found evidence for saccadic suppression already in the retina. *Idrees et al., 2020* demonstrated that retinal ganglion cells in isolated retinae of mice and pigs respond to saccade-like displacements, leading to the suppression of responses to additional flashed visual stimuli through visually triggered retinal-circuit mechanisms. Importantly, their findings suggest that perisaccadic modulations of contrast sensitivity may have a purely visual origin, challenging the need for an efference copy in the early stages of saccadic suppression. However, the suppression they measured lasted much longer than time courses observed in behavioral data. An efference copy signal could thus be necessary to release perception from suppression.

Second, the internal updating of the spatial representation for the saccade vector would be affected. Since intra-saccadic vision is omitted, visual perception is confronted with a retinal image before the saccade and a different one after the performance of the saccade. Predicting the appearance of the post-saccadic image leads to a remapping of internal space and thereby to a seamless perception across the saccade. If the remapping process is not accurate, the appearance of the post-saccadic image would come as a surprise. Surprising stimuli appear much more salient than predicted ones. Failures in trans-saccadic remapping in individuals with autistic symptomatology could be the reason or at least contribute to the sensory overload they experience.

Third, in addition to problems in visual updating, a lack of proper efference copy transfer should produce problems in motor updating. In cases in which visual information is not available or reliable, the sensorimotor system must rely on internal estimates about its own movements.

In the laboratory, trans-saccadic updating can be investigated in motor behavior and in perceptual localization. In the double-step paradigm, two consecutive saccades are made to briefly displayed targets (*Westheimer, 1954*; *Hallett and Lightstone, 1976*). The first saccade occurs without visual references, relying on internal updating to determine the eye's position. The accuracy of the second saccade quantifies the first saccade's updating process.

Perceptual updating is tested similarly, with subjects remembering the position of a target and then indicating it relative to a second target after a single saccade. In this paradigm, one stimulus is shown before and another after saccade execution. Together these two stimuli produce the perception of 'apparent motion'. If stimuli are placed such that the apparent motion path is orthogonal to the saccade path, then the orientation of the apparent motion path indicates how the saccade vector is integrated into vision. The apparent motion trajectory can only appear vertical if the movement of the eyes is perfectly accounted for, that is the retinotopic displacement is largely compensated, ensuring spatial stability. However, small biases of motion direction – implying under- (or over-) compensation of the eye movement – can indicate relative failures in this stabilization process. In a seminal

study, *Szinte and Cavanagh, 2012* found a slight over-compensation of the saccade vector leading to apparent motion slightly tilted against the direction of the saccade. More importantly, when efference copies are not available, that is localization occurring at the time of a second saccade in a double-step task, a strong saccade under-compensation occurs (*Zimmermann et al., 2018*).

This phenomenon cannot be explained by perisaccadic mislocalization of flashed visual stimuli (*Fracasso et al., 2010*; *Szinte and Cavanagh, 2011*), but the two phenomena may be related in that they may both depend upon efference copy information.

In the present study, we directly investigated whether the precision of efference copy signals influences the accumulation of sensory predictions over time and explored potential variations in this mechanism among individuals with a heightened autistic trait. The autistic traits of the whole population form a continuum, with ASD diagnosis usually situated on the high end (*Ruzich et al., 2015*; *Lundström et al., 2012*; *Robinson et al., 2011*). Moreover, autistic traits share a genetic and biological etiology with ASD (*Bralten et al., 2018*). Thus, quantifying autistic-trait-related differences in healthy people can provide unique perspectives as well as a useful surrogate for understanding the symptoms of ASD (*Ruzich et al., 2015*; *Sucksmith et al., 2011*). In two experiments, we focused on the role of efference copy in the pathophysiology of autistic traits, particularly in maintaining perceptual continuity during saccades. We hypothesized that individuals with a high autistic phenotype would exhibit decreased accuracy in second saccades during double-step tasks and under-compensation when localizing a stimulus based on efference copy signals. These findings could shed light on the visual stability and predictive challenges faced by autistic people.

## Results

### Motor updating

We investigated the influence of autistic traits on visual updating during saccadic eye movements using a classic double-step saccade task. This task relies on participants making two consecutive saccades to briefly presented targets. The accuracy of the second saccade serves as an indirect measure of how effectively the participant's brain integrated the execution of the first saccade into their internal representation of visual space. Participants were divided into quartiles based on the severity of their autistic traits, as assessed by the autistic quotient (AQ) questionnaire (*Baron-Cohen et al., 2001*). We hypothesized that individuals with higher autistic traits would exhibit greater difficulty in visual updating compared to those with lower autistic traits. This would be reflected in reduced accuracy of their second saccades in the double-step task. *Figure 2C* illustrates examples from participants at the extremes of the autistic trait distribution (AQ = 3, in orange and AQ = 31, in magenta). As shown, both participants were instructed to make saccades to the locations indicated by two brief target appearances (T1 and T2), as quickly and accurately as possible, following the order of presentation. However, successful execution of the second saccade requires accurate internal compensation for the first saccade, without any visual references or feedback available during the saccade itself.

To control for general impairments in saccade generation, we first checked for the accuracy of the first saccade (from FP to T1). *Figure 3A* shows the average saccade landing positions of the two subgroups of participants (lower and upper quartiles of the AQ distribution in orange and magenta, respectively) in in the double-step task. For the analysis, we collapsed data from trials with second saccades into the lower and upper visual field. For statistical analyses, we first compared horizontal landing positions of the first saccade for the two groups (*Figure 3B*). Saccade landing positions of participants in the lower quartile (mean degree ± standard error of the mean [SEM]: 10.17 ± 0.50) did not deviate significantly from those in the upper quartile (mean degree ± SEM: 9.65 ± 0.77). This result was also confirmed by a two sample $t$-test ($t(14) = 0.66$; $p = 0.66$, BF10 = 0.40) and by a non-significant correlation between the landing position of the first saccade and the autistic traits of our participants ($r = -0.09$; $p = 0.55$, BF10 = 0.14, not shown as a figure).

We then calculated the angle between the participants' second saccade vector and the optimal vector that would have led to the physical target position T2 from the second saccades' starting position. The analysis of the second saccade provided the means to determine the effect of efference copies signals in underlying extra-retinal information. If the participant fails to compensate for the first saccade due to disrupted efference copy, the second saccade vector deviates from the location of the second target (*Figure 1B*). A correct remapping of the previously seen target position would lead

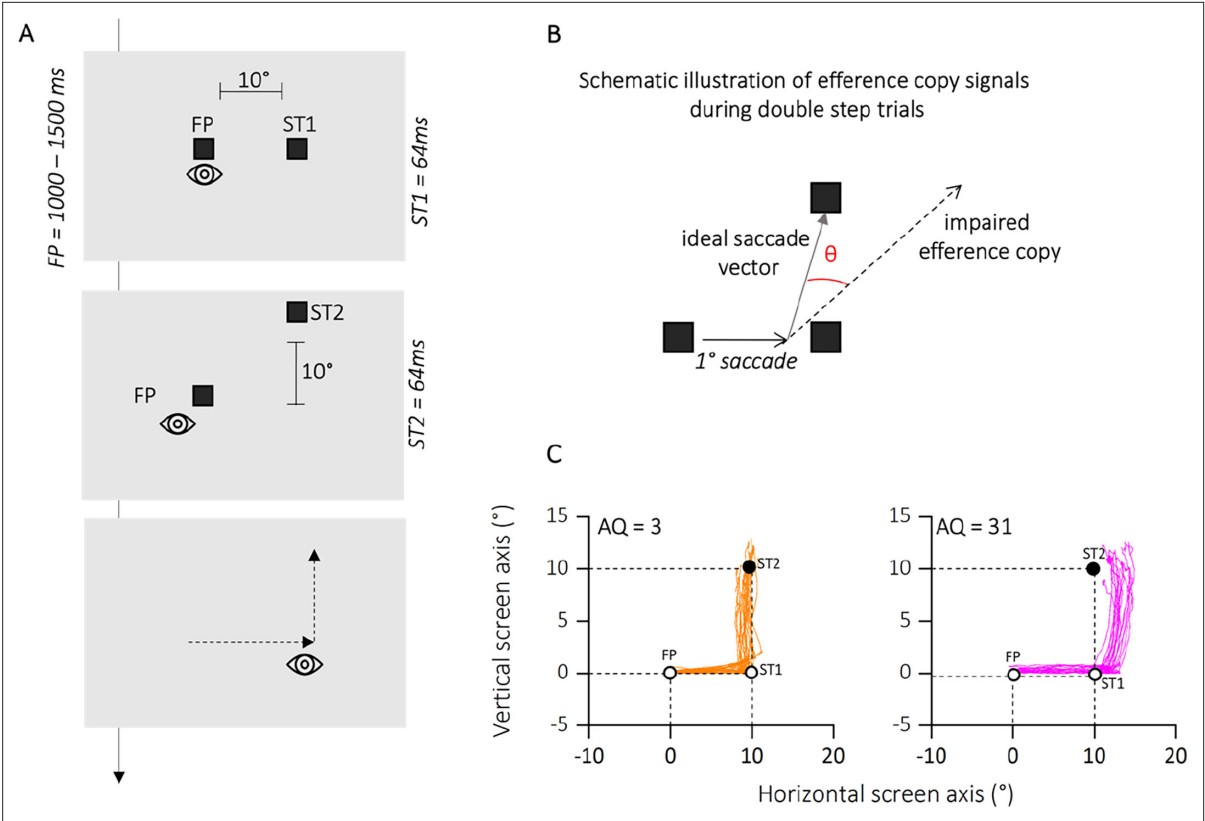

**Figure 2.** Methods experiment 1. (**A**) Time course of presentations and spatial arrangement of stimuli during double-step task. Participants made double-step saccades from a fixation point (FP) to the first saccadic target (ST1), and from there to a second saccade target (ST2). Both FP and saccade targets were separated by 10°. Participants were instructed to initiate the saccade only after all stimuli had disappeared, so to avoid any visual references. (**B**) Hypothetical pattern of first and second saccades on a particular trial and illustration of the variables angle shift and horizontal endpoint shift of second saccades. We determined how far the second saccade deviated from the optimal vector that would have directed the gaze onto the target. We calculated the angle between the second saccade vector and the optimal vector connecting the starting position of the second saccade and the second saccade target. These angles were computed separately for upward and downward saccades, but since they resulted in very similar values, we collapsed the data into upward angles only. (**C**) Eye position traces for two representative participants. The eye position between the initiation of the first saccade from FP toward ST1 (empty circles) and the termination of the second saccade toward ST2 (filled circle) are depicted for each trial in which successive saccades to ST1 and ST2 were produced. Plots are separated for low autistic traits (autistic quotient [AQ] = 3, in orange) and high autistic traits (AQ = 31, in magenta) example participants.

to smaller angle of deviations from the optimal saccade vector; however, a failure in using efference copy-based updating of visual space would results in larger deviations. *Figure 3C* plots the motor bias (expressed as angle of deviations) as a function of autistic traits severity. We found a significant correlation between the autistic characteristics of our group of participants and the amount of motor bias ($r = 0.52$; $p < 0.001$; BF10 = 57.37). This result is also confirmed by a strong group difference, between the lower and upper quartile of AQ distribution ($t(14) = -4.68$; $p = 0.002$; BF10 = 21.45), with a mean bias of 6.26 ± 0.80 for the lower quartile and of 20.61 ± 3.40 for the upper quartile.

Results on the double-step task suggest that although participants did not show an impairment in the first saccade execution, those with higher autistic traits reported difficulties in using extra-retinal information about the amplitude and direction of the motor vector of the first saccade, in order to update the spatial representation of the second target. Thus, they were less able to use the efference copy of the first saccade to construct a spatial representation of the second target location in generating an accurate saccade toward it.

## Visual updating

With a trans-saccadic localization task, we explored how autistic traits affect the integration of eye movements into visual perception. Participants were presented with stimuli before and after a

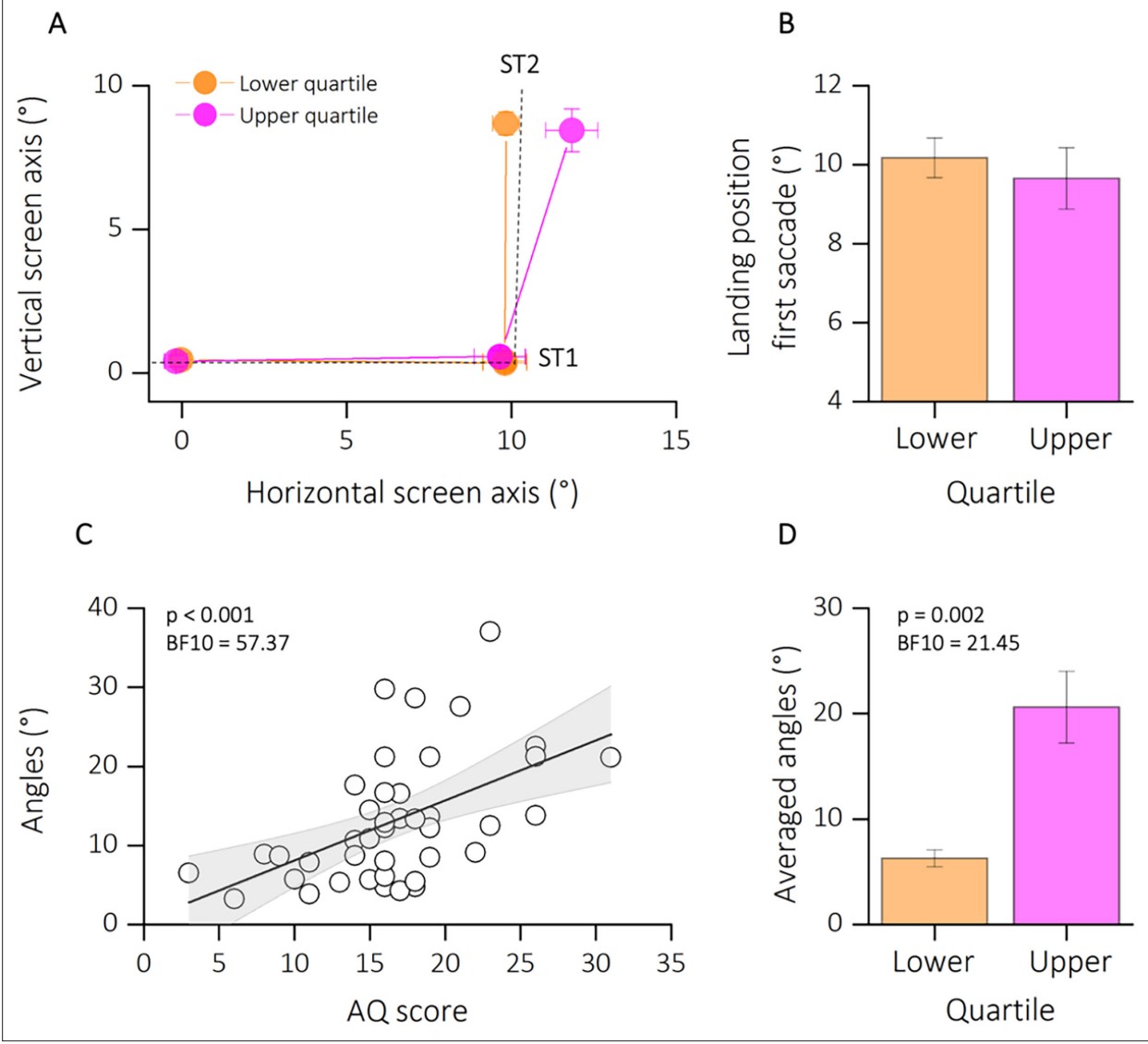

**Figure 3.** Results experiment 1. (**A**) Average saccade vectors from FP (0, 0) to ST1 (10, 0) and from ST1 to ST2 (10, 10). Data are shows for a subsample of participants, falling in the lower (orange) and upper (magenta) quartiles of the autistic quotient (AQ) distribution. Data from upward and downward second saccades are collapsed. Dashed lines indicate the required saccade path from FP to ST1 and from ST1 to ST2. (**B**) Average first saccade landing position for lower and upper quartiles of the AQ distribution. Error bars are standard error of the mean (SEM). (**C**) Linear regression between angles (distance the actual second saccade vector and the optimal vector that would have led to the physical target position ST2 from the second saccades' starting position) and the autistic score of participants. Text insets report p-values and associated Bayes Factors of Pearson's Rho. Thick black line shows the linear fit through the data. The shaded area represents the standard error for each observation mean as predicted by the regression line. (**D**) Average angles (°) for lower (AQ <13, orange) and upper (AQ >19, magenta) quartiles of the AQ distribution. Error bars are SEM. Text insets report p-values and associated Bayes Factors of paired sample *t*-test.

single saccade, creating an illusion of apparent motion. We measured the perceived direction of this displacement, which is influenced by how well the participant's brain accounts for the saccadic eye movement. We predicted that individuals with higher autistic traits would show a stronger bias in the perceived displacement direction, suggesting a less accurate integration of the eye movement into their visual perception.

Participants were instructed to report the position of a second dot relative to a first dot, after saccade execution (see *Figure 4A*). The two dot stimuli, presented below and above the saccade path, were horizontally displaced from each other to a variable degree. One dot was always presented before saccade onset, the other after saccade completion, so that the saccade dissociates retinal from screen coordinates. To perceive vertical trajectory in screen coordinates, the saccade vector

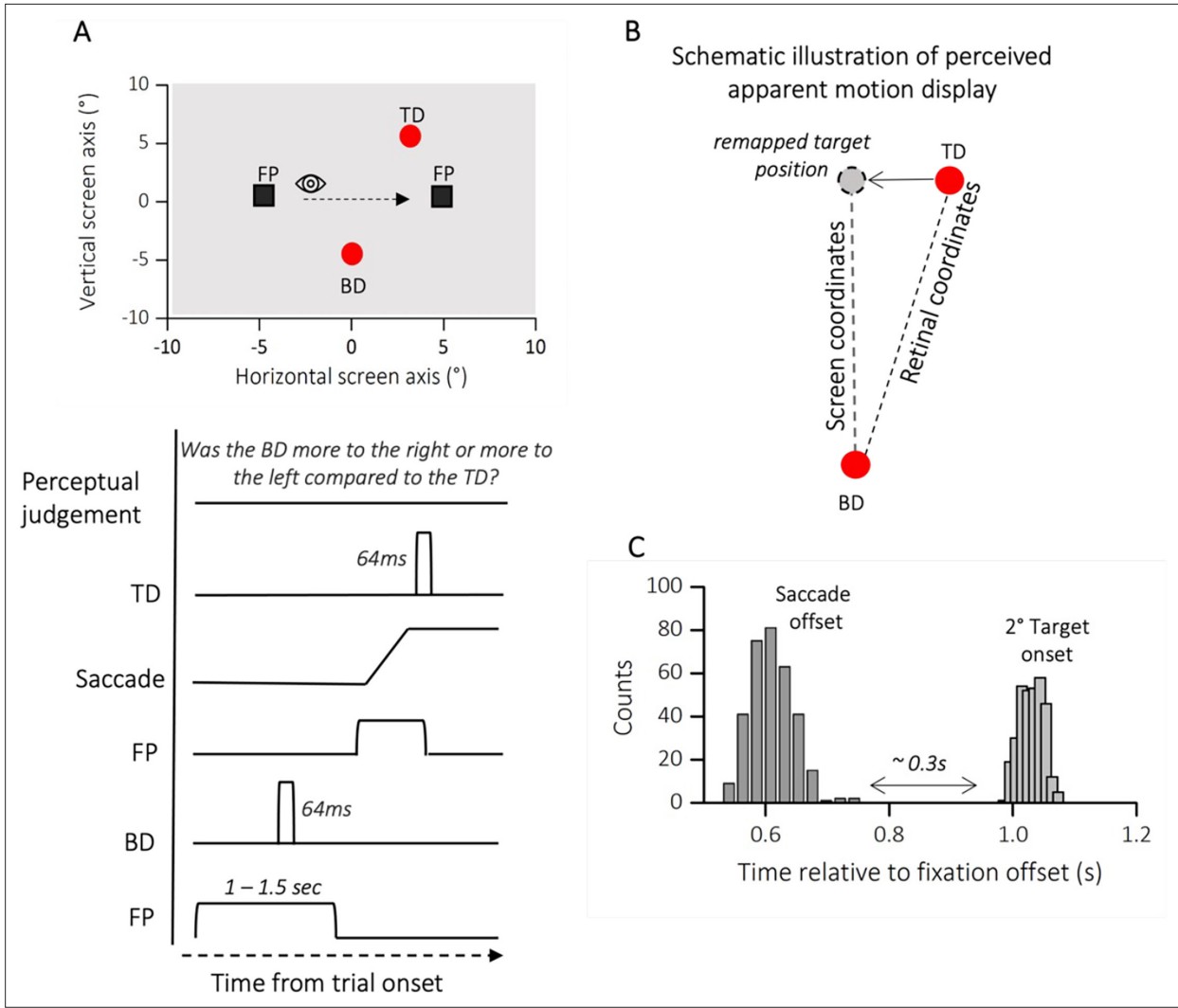

**Figure 4.** Methods experiment 2. (**A**) Spatial arrangement of stimuli and time course of presentations during trans-saccadic localization task. Participants made a saccade from a fixation point (FP, left) to a second fixation point (FP, right). At a random time before the first FP disappearance, a red circle (Bottom Dot, BD) was presented for 64 ms, at saccade completion a second red dot (Top Dot, TD) was delivered for another 64 ms. Participants were instructed to initiate the saccade only after the first FP had disappeared. At saccade completion participants were asked to report whether the first dot delivered (in the figure BD) was more to the right or more to the left compared to the second one (in the figure TD). (**B**) Schematic illustration of perceived apparent motion display. On the retina, the first probe falls to the right side of the fovea while the second falls to the left side. To compensate for these effects of the saccade, the visual system corrects the expected location of the second dot in the opposite direction to the saccade (remapped target position, gray dot). If this correction is accurate, the displacement is perceived in its spatiotopic (vertical dashed line: from remapped target position to BD) rather than retinotopic arrangement (oblique dashed line: from TD to BD) and space constancy is maintained. (**C**) Average frequency distribution of saccade offsets and second target onset. We analyzed only the trials in which the second target was delivered after saccade completion (on average around 0.3 s), so that the visual system could dissociate the retinal from the screen coordinates.

must be compensated for – likely through efference copy information. At saccade completion, on the retina the first probe falls to the right side of the fovea while the second falls to the left side (see *Figure 4B*, retinal coordinates). To compensate for the effects of the saccade, the visual system corrects the expected location of T2 in the opposite direction to the saccade (black arrow, remapped target position). If this correction is accurate, the displacement of the dots is perceived relatively vertical (*Figure 4B*, screen coordinates), rather than in retinotopic direction (relatively oblique, gray dashed line) and space constancy is maintained. Every shift from the vertical trajectory indicates an under- or over-compensation of the saccade vector. Previous studies have shown that the dots trajectory is indeed seen as approximately vertical, although a systematic slant of trajectories implies a tendency to over-compensate for the saccade vector (*Szinte and Cavanagh, 2012*; *Hui et al., 2020*).

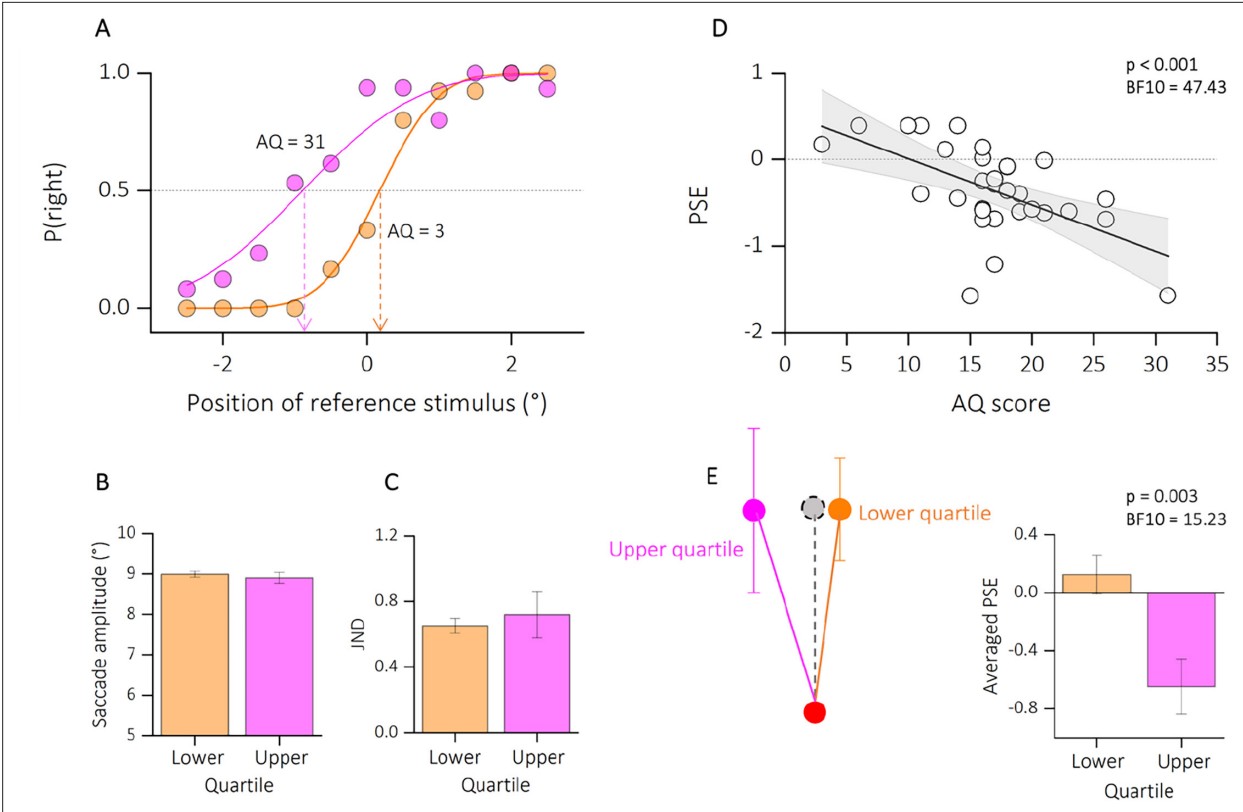

**Figure 5.** Results experiment 2. (**A**) Psychometric functions for two example participants at the very extremes of the autistic quotient distribution (AQ = 3, in orange; AQ = 31, in magenta).The functions plot the proportion of reporting 'right', as function of the position of the second dot relative to the first one (shown in the abscissa). The vertical-colored lines show the estimates of the PSE, given by the median of the fitted cumulative Gaussian functions. (**B**) Average saccade amplitude (°) for lower (AQ <14, in orange) and upper (AQ >20, in magenta) quartiles of the autism distribution. Saccade amplitude did not vary with autistic traits. (**C**) Precision thresholds (Just Noticeble Difference, JND) for lower and upper quartiles of the AQ distribution, same convention as in B. Here, the precision in reporting the dot displacement was similar between subsamples. (**D**) Linear regression between PSE and the autistic score of participants. Text insets report p-values and associated Bayes Factors of Pearson's Rho. Thick black line shows the linear fit through the data. The shaded area represents the standard error for each observation mean as predicted by the regression line. (**E**) Schematic illustration of perceived dots displacement for upper and lower quartiles of the AQ. A localization compensating fully for the saccadic eye movement would lead to a perfect vertical localization (gray dot, dashed line). However, deviations from the vertical indicate and under- or over-compensation of the saccade vector. We found a large under compensation of the saccade vector for the participants with higher autistic traits, compared to participants with lower autistic traits. See also bar plot, average PSE, on the right. Text insets report p-values and associated Bayes Factors of two sample *t*-test.

However, if the retinotopic displacement is uncompensated, a failure in spatial stability is encountered. *Figure 5A* plots two example psychometric functions for one participant at the extremes of the lower (AQ = 3, in orange) and upper quartiles of the distribution (AQ = 31, in magenta). The mean of the psychometric function (PSE) represented the location at which the references and the probe appeared to be vertically aligned. The PSE for the two example participants resulted very different: while ~0° bias represents accurate, spatiotopic correction, a negative bias (PSE <0) reflects an under compensation of the saccade vector.

We first checked that all participants were able to execute accurate saccades. The saccade amplitude did not correlate with the autistic traits of our participants (*r* = −0.18; p = 0.33; BF10 = 0.22, not shown as a figure). Moreover, when checking on the mean amplitude across trials (*Figure 5B*), we did not find a significant difference between the lower (mean degree ± SEM: 8.66 ± 0.27) and upper (mean degree ± SEM: 7.96 ± 0.46) quartiles of participants AQ distribution (two sample *t*-test, t(13) = 1.42; p = 0.17; BF10 = 0.62).

We then checked that all participants were able to complete the task. A measure of precision (JND) was extracted from each participant's psychometric function. *Figure 5C* shows the averaged JND for lower and upper quartiles of the distribution. We report a non-significant difference between the upper and lower quartiles of the autistic score's distribution (t(13) = −0.46; p = 0.64; BF10 = 0.29) nor

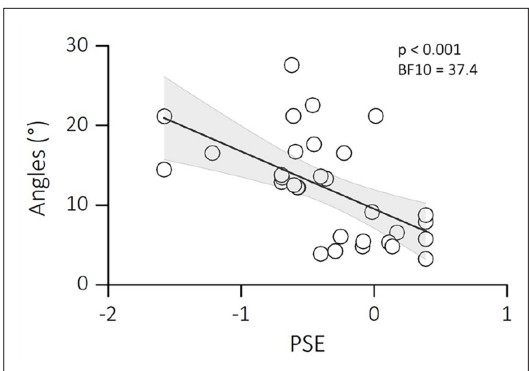

**Figure 6.** Linear regression between PSEs of the trans-saccadic localization task and the angles of deviation from the double-step saccade task. Text insets report p-values and associated Bayes Factors of Pearson's Rho. Thick black line shows the linear fit through the data. The shaded area represents the standard error for each observation mean as predicted by the regression line.

a significant correlation between JND and the AQ of all our participants ($r = 0.17$; $p = 0.36$; BF10 = 0.21, not shown as a figure). Indeed, the averaged precision in reporting the position of the second dot was very similar between the upper (mean ± SEM: 0.72 ± 0.04) and lower (mean ± SEM: 0.65 ± 0.14) quartiles of the distribution.

In contrast, the averaged biases (expressed as the PSE) varied as a function of the autistic trait severity (*Figure 5D*): the higher the autistic scores the stronger the bias in reporting the displacement in retinotopic coordinates, hence correcting less for the saccade eye movement ($r = -0.58$; $p < 0.001$; BF10 = 47.43).

Indeed, while we report a slightly tilt of the dots trajectory for the lower quartile of the distribution (mean ± SEM: 0.12 ± 0.13), the reported bias was tilted in the opposite direction for participants falling in the upper quartile (mean ± SEM: −0.64 ± 0.19), implying an under-compensation of the saccade vector and a partial failure in visual stability (see *Figure 5E* for a schematic illustration of the results).

This result is also confirmed by a significant difference between the two groups ($t(13) = 3.63$; $p = 0.003$; BF10 = 15.23; *Figure 4E*, bar plot).

We finally examined the relationship between the motor and visual biases in the participants that completed both tasks. *Figure 6* plots the angles of deviations (motor bias) as a function of the PSE of our participants (visual bias). We found that motor and visual biases very strongly correlated ($r = -0.57$; $p < 0.001$; BF10 = 37.4), suggesting that a similar mechanism – controlled by the same efference copy signal – might be responsible for both motor and visual updating.

Understanding how these biases might change over time could provide further insights into this mechanism. Specifically, we investigated whether participants exhibited any learning effects throughout the experiments. For data of Experiment 1 – motor updating – we divided our data into 10 separate bins of 30 trials each. We conducted a repeated measure analysis of variance (ANOVA) with the within-subject factor 'number of sessions' (two main sessions of 5 bins each, ~150 trials) and the between-subject factor 'group' (lower vs upper quartile of the AQ distribution). We found no main effect of 'number of sessions' ($F(1,7) = 0.25$, $p = 0.66$), a main effect of 'group' ($F(1,7) = 2.52$, $p = 0.015$), and no interaction between the two subsample of participants and the sessions tested ($F(1,7) = 0.51$, $p = 0.49$). Data of Experiment 2 – visual updating– were separated into three sessions. For each session, we extracted the PSE and we conducted a repeated measure ANOVA with within-subject factor 'sessions' and between-subject factor 'groups' (lower vs upper quartile of the AQ distribution). Also here we found no main effect of sessions ($F(1,13) = 0.86$, $p = 0.39$), a main effect of group ($F(1,14) = 11.85$, $p = 0.004$), and no interaction between the two subsample of participants and the sessions tested ($F(1,13) = 0.20$, $p = 0.73$). In conclusion, the current study found no evidence of learning effects across the experimental sessions. However, a significant main effect of group was observed in both Experiment 1 (motor updating) and Experiment 2 (visual updating). Participants in the group with higher autistic traits performed systematically differently on the task, regardless of the number of sessions completed compared to those in the group with lower autistic traits.

## Discussion

In the current study, we examined oculomotor efference copy signals in a sample of healthy adults with variable autistic traits using both, a motor and visual updating task. We show that a disrupted efference copy mechanism can lead to inaccurate predictions about impending eye movements,

forcing the visuo-motor system to rely more heavily on actual sensory input rather than the anticipated sensory consequences of self-generated eye movements.

In our first experiment, the double-step saccade task, we found an impairment in the capacity of participants exhibiting high levels of autistic traits to program double-step saccades which require efference copy information about the size of the previous saccade. Conversely, participants with relatively low autistic characteristic perform accurately in the double-step task. In the high-autism-like subsample, the final eye position after the second saccade markedly deviates from the position of the second target, and the magnitude of this error far surpasses that observed in the low-autism-like group. This specific impairment in the accuracy of second saccades lends support to the hypothesis that heightened autistic traits correlates with a deficit in constructing a spatial representation of the second target's location based on forward models of efference copy. Notably, the mislocalization of the second saccade target cannot be explained by differences in the dynamics of the saccade to the first target: The observation that the accuracy of the initial saccade in the double-step task remains consistent regardless of the degree of autism-like trait severity implies that individuals with more pronounced traits do not exhibit inherent challenges in the encoding and processing of spatial information for intended saccades. Instead, when preparing for a saccade that must compensate for a preceding eye movement that altered the retinotopic representation of the target, those with more severe traits encounter difficulties. Likewise, the bias in localizing the target cannot be attributed to working memory deficits within our high-autism-like-symptom subsample; it is improbable that the mislocalization of the second target results solely from documented working memory deficits in ASD (*Han et al., 2022*).

Our findings suggest that a general memory deficit is unlikely to fully explain the observed bias in high-AQ participants' second saccades. As highlighted in *Figure 3A*, the bias was specific to the horizontal dimension, weakening the argument for a global memory issue affecting both vertical and horizontal encoding of target location. A *t*-test between our group of participants revealed a difference in saccade accuracy for the *x* dimension (p = 0.03) but not in the *y* dimension (p = 0.88). However, it is important to acknowledge that even under non-darkness conditions, participants might rely on a combination of internal updating based on the initial target location and visual cues from the environment, such as screen borders. This potential use of visual references could contribute to the observed bias in the high-AQ group. If high-AQ participants differed in their reliance on visual cues compared to the low-AQ group, it could explain the specific pattern of altered remapping observed in the horizontal dimension. This possibility aligns with our argument for an abnormal remapping process underlying the results. While altered efference copy signals remain a strong candidate, the potential influence of visual cues on remapping in this population warrants further investigation. Future studies could incorporate a darkness condition to isolate the effects of internal updating on the first saccade, and systematically manipulate the availability of visual cues throughout the task. This would allow for a more nuanced understanding of how internal updating and visual reference use interact in the double-step paradigm, particularly for individuals with varying AQ scores.

Autistic people show remarkable similarities to the symptoms of schizophrenia patients (*Barlati et al., 2020*; *Hommer and Swedo, 2015*; *De Crescenzo et al., 2019*; *Krieger et al., 2021*). Both ASD and schizophrenia patients are considered to have impairments at both extremes, involving either an excessive or inadequate reliance on sensory signals, resulting in inflexible behavior and issues in recurrent neural network representation (*Idei et al., 2017*; *Philippsen and Nagai, 2018*). Using a double-step saccade task, previous research has reported multiple lines of evidence demonstrating disrupted efference copy signals in schizophrenia. Importantly, similar to our findings, this disruption has implications for perceptual and motor precision, highlighting the interconnectedness of corollary discharge deficits and motor inflexibility (*Thakkar et al., 2015b*; *Thakkar et al., 2015a*).

A dramatic consequence of altered efference copy signaling is the inability to account for own eye movements when judging the location of a target presented intra-saccadically. The trans-saccadic localization task yielded results that provide further insights into the role of efference copy in individuals with varying levels of autistic traits. In this task, all participants, regardless of their autistic trait levels, successfully executed accurate saccades to peripheral targets. However, the key factor emerged in the subsequent perceptual judgments made by these individuals. Individuals with high autistic traits exhibited a significant mislocalization of a remembered stimulus position across saccades. In essence, they perceived the visual stimulus as having shifted in space more than it actually did during the eye

movement. This indicates that their brain is relying more on the actual sensory feedback than the efference copy prediction. The consequence is an inaccurate perception of the stimulus' location in space.

The results of both experiments shed light on the relationship between efference copy failure and Bayesian theories of autism (*Van de Cruys et al., 2014*; *Pellicano and Burr, 2012*; *Lawson et al., 2014*). In the double-step task, individuals with higher autistic traits exhibited difficulties in utilizing efference copy signals to construct a spatial representation of the second target location. This impairment suggests that in high autistic characteristics pre-saccadic predictions will never be completely confirmed by the post-saccadic feedback. This constant lack of a successful match between predicted sensory consequences of self-generated eye movements and post-saccadic evidence can lead over time to the build-up of weak priors with high variance.

Similarly, in the trans-saccadic localization task, high autistic traits were associated with failures in spatial updating across saccade execution. These findings align with the Bayesian framework, where the brain continuously updates its internal model by combining sensory information with prior knowledge. In the context of autism, a failure in efference copy can further contribute to the unique sensory integration processes observed in individuals on the autism spectrum. Our results are supported by recent evidence on the clinical population. Using a blanking task in a group of children with and without Autism (*Yao et al., 2021*) suggested that that the ability to flexibly adjust the precision of priors may serve a protective function in autistic people, potentially influencing the severity of symptom presentation. Moreover, *Park et al., 2021* highlighted a unique relationship between smooth pursuit quality and motion prediction in ASD: unlike typically developing children, those with ASD did not exhibit a central-tendency bias over time, suggesting deficits in integrating prior knowledge about environmental statistics.

We have recently suggested a disruption in causal inference in the high autistic characteristics, as these individuals tend to attribute post-saccadic errors to external factors – a dynamic and unstable world – rather than accurately linking them to their own motor actions (*Pomè et al., 2023*). Consequently, a bias in causal inference emerges, stemming from the efference copy disruption, where the brain struggles to correctly attribute the cause of motor errors, leading to a heightened reliance on sensory feedback instead of motor predictions. As previously postulated this skewed causal inference also interconnect with motor inflexibility. Inflexibility in the motor domain might also have consequences for visual stability. Here, participants with high autistic traits faced challenges in executing saccades with precision, particularly when compensating for previous eye movements that had altered target representations. In the trans-saccadic localization task, we observed that these individuals leaned more heavily on post-saccadic sensory feedback than efference copy predictions. This not only led to an inaccurate perception of stimuli location but also underscored their limited ability to update and adapt motor actions based on sensory information.

It is essential to note that our participant pool lacked pre-existing diagnoses before engaging in the experiments and we must address limitations associated with the AQ questionnaire. The AQ questionnaire demonstrates adequate test–retest reliability (*Baron-Cohen et al., 2001*), normal distribution of sum scores in the general population (*Hurst et al., 2007*), and cross-cultural equivalence has been established in Dutch and Japanese samples (*Hoekstra et al., 2008*; *Kurita et al., 2005*; *Wakabayashi et al., 2006*). The AQ effectively categorizes individuals into low, average, and high degrees of autistic traits, demonstrating sensitivity for both group and individual assessments (*Tennant and Conaghan, 2007*).

However, evolving research underscores many aspects that are not fully captured by the self-administered questionnaire: for example, gender differences in ASD trait manifestation (*Lai et al., 2011*). Autistic females may exhibit more socially typical interests, often overlooked by professionals (*Simcoe et al., 2023*). Camouflaging behaviors, employed by autistic women to blend in, pose challenges for accurate diagnosis (*Cook et al., 2022*). Late diagnoses are attributed to a lack of awareness, gendered traits, and outdated assessment tools (*Zener, 2019*). Moving forward, complementing AQ evaluations in the general population with other questionnaires, such as those assessing camouflaging abilities (*Hull et al., 2019*), or motor skills in everyday situation (MOSES-test *Hillus et al., 2019*) becomes crucial for a comprehensive understanding of autistic traits.

In conclusion, our study underscores the critical role of efference copy in maintaining perceptual stability during eye movements and highlights its potential relevance to the sensory processing

differences observed in individuals with heightened autistic traits. These findings contribute to our understanding of the complex interplay between prior knowledge, sensory integration, and motor control in the context of ASD, opening avenues for future research and potential interventions aimed at enhancing perceptual stability in this population.

# Methods

**Key resources table**

| Reagent type (species) or resource | Designation | Source or reference | Identifiers | Additional information |
|---|---|---|---|---|
| Software, algorithm | MATLAB code to process data | This paper and Mathworks https://mathworks.com | RRID:SCR_001622 | |

## Participants

Forty-two participants (31 females, mean age 23, SD = 4.19) took part to Experiment 1 (motor updating). Thirty-one of them (21 females, mean age 22.81, SD = 3.11) participated also in Experiment 2 (visual updating). Subjects were either German native speakers or English speakers with no neurological or psychiatric diseases. Participants either reported to have normal vision or they wore lenses during their acquisition. All participants were recruited through the Heinrich-Heine University Düsseldorf and received either course credit or payment of 10 euros per hour. The experimental procedure was approved by the local ethics committee of the Faculty of Mathematics and Natural Sciences of Heinrich-Heine-University, Düsseldorf (ethics approval number: PO01_2022_01). The research was in accordance with the Declaration of Helsinki and informed consent was obtained from all participants prior to the experiment.

## Experimental materials and procedures

Stimuli were displayed on a 27-inch Acer XB272 LCD monitor driven by a Alienware pc (Aurora R7) with a 240-Hz refresh rate (frame duration 4.16 ms) and a resolution of 1920 × 1080 pixels. The experimental program was implemented in MATLAB 2016b (Mathworks, Natick, MA, USA) using Psychtoolbox (*Kleiner et al., 2007*). Eye movements and pupil diameters were recorded with the EyeLink 1000 system (SR Research Ltd, Mississauga, Ontario, Canada), which sampled eye positions at a rate of 1000 Hz. The head was sustained with a chin- and forehead-rest. For all participants, the left eye was recorded. Viewing was binocular. At the beginning of each session, the Eyelink was calibrated with the standard nine-point Eyelink procedure. The system detected the start and the end of a saccade when eye velocity exceeded or fell below 30°/s.

## Experiment 1: motor updating

A trial started with the presentation of a black fixation point (0.55 × 0.55°, FP) at the screen center. After 1000 ms plus a randomly chosen duration between 0 and 500 ms, the first saccade target ST1 (0.55 × 0.55°, black) appeared 10° to the right of the screen center (see *Figure 2A*), which remained visible for 64 ms. Upon extinction of ST1, ST2 appeared 10° upwards or downwards from ST1, and remained visible for another 64 ms. The position of ST2 was randomized across trials. ST2 and the fixation point were extinguished together, and these cued participants to start the saccade sequence: from the fixation point to ST1, and from ST1 to ST2. Participants completed 3 sessions of 100 trials each.

## Experiment 2: visual updating

The trial sequence is shown in *Figures 3B and 4A*. A trial started with the presentation of a fixation point (black square FP, 0.55° × 0.55°) 5° to the left of the screen center. The fixation point stayed on for 1000 ms plus a randomly chosen duration between 0 and 500 ms. At a random time within the fixation point presentation time, the first stimulus was presented (Bottom Dot, BD). After disappearance of the first fixation point, participants were instructed to saccade to the second fixation point (black square FP, 0.55° × 0.55°) 5° to the right of the screen center (total saccade size 10°). At saccade completion, a second stimulus was delivered (Top Dot, TD). The stimuli consisted of two red dots (1.5° diameter), each flashed for one monitor frame with at least a temporal separation of 300 ms on average (see

*Figure 4C* for distribution of saccade offset and second stimulus onset). The first dot could appear randomly above or below gaze level at a fixed horizontal location, halfway between the two fixations (x = 0, y = −5° or +5° depending on the trial). The second dot was then shown orthogonal to the first one at a variable horizontal location (x = 0° ± 2.5°). The stimulus was perceived as a single dot moving downward or upward (depending on the position of the first dot) with a near-vertical trajectory, that is orthogonal to the direction of saccades. Participants completed 3 sessions of 110 trials each.

## Quantification and statistical analysis

### Experiment 1

In the analysis of eye movement data in Experiment 1 we excluded trials if: (1) participants did not perform a saccade or they blinked during the execution of the saccade; (2) participants started the saccade sequence before the extinction of the fixation point; (3) the amplitude of the first saccade was smaller than half of the required distance, that is <5°; (4) the vertical amplitude of the second saccade was smaller than 5°; (5) participants deviated more than 2.5° on the horizontal or vertical dimension when fixating; (6) the latency of the first saccade was <100 ms. Saccade amplitudes were averaged into 10 separate bins of ~30 trials each for first, and second saccades that passed the selection criteria. These criteria were applied to ensure that both saccades were large enough to reveal a putative deviation of the second saccade. If, for instance, the executed first saccade is much smaller than the required distance, the efference copy should signal a smaller amplitude, thus leading to smaller influences on the direction of the second saccade. We tested our hypothesis concerning impaired use of efference copy information in high autism phenotype by analyzing the direction of the second saccade. Specifically, we determined how far the second saccade deviated from the optimal vector that would have directed the gaze onto the target. We calculated the angle between the second saccade vector and the optimal vector connecting the starting position of the second saccade and the second saccade target (see *Figure 2B*). These angles were computed separately for upward and downward saccades, but since they resulted in very similar values, we collapsed the data into upward angles only.

### Experiment 2

We only analyzed trials where the saccade was performed between the presentation of the two dots (first dot at least 60 ms before the saccade, second dot after its completion), therefore, in retinotopic coordinates, the two dots were always displaced horizontally by about 10°. That is when the dots were presented on the screen, they appeared at a different location on the retina, horizontally separated by an angle of approximatively 10° (see also *Figure 4B*). In other words, each dot was perceived on the retina at a slightly different position in the visual field, due to horizontal displacement. That subjects perceived the dots displaced along a nearly vertical trajectory, indicates that the retinotopic displacement is largely compensated, ensuring spatial stability. However, small biases of reported direction can indicate relative failures in this stabilization process. To estimate biases in trans-saccadic updating, we varied the location of the second dot with the method of constant stimuli and asked subjects to report in 2AFC whether the second dot was more slanted to the right or to the left compared to the first one. Data were analyzed as psychometric functions, plotting the proportion of 'rightward' judgments as function of the position of the second dot relative to the first one. Distributions were fitted with cumulative Gaussian functions; the median of the curve estimated the PSE, or the position of the second dot that led to vertical stimulus displacement. A negative bias (PSE <0) implies a bias toward seeing rightward displacement and a positive bias (PSE >0) implies a bias toward seeing leftward displacement (as in *Figure 3B*, dashed gray line). The former negative bias can be interpreted as an under-compensation of the saccade vector. Moreover, a trial was discarded if: (1) participants did not perform a saccade or they blinked during the execution of the saccade; (2) participants started the saccade before the extinction of the fixation point; (3) the amplitude was smaller than half of the required distance; (4) participants deviated more than 2.5° on the horizontal or vertical dimension when fixating.

## Autistic quotient

All participants completed the self-administered AQ questionnaire, in the German or English validated version (*Ruzich et al., 2015*; *Freitag et al., 2007*). This contains 50 items, grouped in five

subscales: attention switching, attention to detail, imagination, communication, and social skills. For each question, participants read a statement and selected the degree to which the statement best described them: 'strongly agree', 'slightly agree', 'slightly disagree', and 'strongly disagree'. The standard scoring described in the original paper was used: 1 when the participant's response was characteristic of ASD (slightly or strongly), 0 otherwise. Total scores ranged between 0 and 50, with higher scores indicating higher degrees of autistic traits. All participants scored below 32, the threshold above which a clinical assessment is recommended (*Baron-Cohen et al., 2001*).

The median of the scores for the first experiment was 16, with lower and upper quartiles of 13 and 19. Scores were normally distributed, as measured by the Jarque–Bera goodness-of-fit test of composite normality (JB = 0.98, p = 0.5). The median score of the subsample participants for Experiment 2 was 17, with lower and upper quartiles of 14 and 20. Scores were normally distributed, as measured by the Jarque–Bera goodness-of-fit test of composite normality (JB = 1.39, p = 0.31).

As we were interested in the effect of autistic personality traits on the results, correlation analyses were complemented with standard *t*-tests comparing the upper and lower quartiles of the AQ distribution's scores for the two experiments separately. Both measures were accompanied with Bayes Factors estimation. Bayes Factors (*Rouder et al., 2009*) quantify the evidence for or against the null hypothesis as the ratio of the likelihoods for the experimental and the null hypothesis. We express it as the ratio, where negative numbers indicate that the null hypothesis is likely to be true, positive that it is more likely false. By convention, absolute Bayes Factors >3 are considered substantial evidence for either the alternate or null hypothesis, >10 strong evidence, and >100 decisive.

## Acknowledgements

AP discloses support for the research of this work from the European Union's Horizon 2020 research and innovation program under the Marie Skłodowska-Curie (Grant Agreement Number 101029574–APPROVE 'Autistic Perception and the predictive role of visual experience'). EZ discloses support for publication of this work from the German Research Foundation (Deutsche Forschungsgemeinschaft; DFG; Grant Agreement Number ZI 1456/6-1) and from the European Research Council (ERC) under the European Union's Horizon 2020 research and innovation programs (Grant Agreement Number 757184–moreSense). We thank Leonie Otto for the data collection.

## Additional information

### Funding

| Funder | Grant reference number | Author |
| --- | --- | --- |
| European Research Council | 757184 | Eckart Zimmermann |
| Marię Sklodowska Curie | 101029574 | Antonella Pomè |
| Deutsche Forschungsgemeinschaft | ZI 1456/6-1 | Eckart Zimmermann |

The funders had no role in study design, data collection, and interpretation, or the decision to submit the work for publication.

### Author contributions

Antonella Pomè, Conceptualization, Data curation, Formal analysis, Funding acquisition, Investigation, Visualization, Methodology, Writing – original draft, Project administration, Writing – review and editing; Eckart Zimmermann, Supervision, Validation, Writing – review and editing

### Author ORCIDs

Antonella Pomè [iD] https://orcid.org/0000-0003-1116-5399

## Ethics

The experimental procedure was approved by the local ethics committee of the Faculty of Mathematics and Natural Sciences of Heinrich-Heine-University, Düsseldorf (ethics approval number: PO01_2022_01).

Reviewer #1 (Public Review): https://doi.org/10.7554/eLife.94946.3.sa1
Reviewer #2 (Public Review): https://doi.org/10.7554/eLife.94946.3.sa2
Reviewer #3 (Public Review): https://doi.org/10.7554/eLife.94946.3.sa3
Author response https://doi.org/10.7554/eLife.94946.3.sa4

## Additional files

### Supplementary files

• MDAR checklist

### Data availability

Data has been deposited in Zenodo (https://doi.org/10.5281/zenodo.11277648).

The following dataset was generated:

| Author(s) | Year | Dataset title | Dataset URL | Database and Identifier |
|---|---|---|---|---|
| Pomè A, Zimmermann E | 2024 | Visuo-motor updating in individuals with heightened autistic traits | https://zenodo.org/records/11277648 | Zenodo, 10.5281/zenodo.11277648 |

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
