## [Editor Report · eLife assessment]

This **important** study shows that a high autism quotient in neurotypical adults is associated with suboptimal motor planning and visual updating after eye movements, suggesting a disrupted efference copy mechanism. The implication is that abnormal visuomotor updating may contribute to sensory overload - a key symptom in autism spectrum disorder. The evidence presented is **convincing**, with few limitations, and should be of broad interest to neuroscientists at large.

---

## [Referee Report · Reviewer #1 (Public Review)]

Summary:

This study examines a hypothesized link between autism symptomatology and efference copy mechanisms. This is an important question for a number of reasons. Efference copy is both a critical brain mechanism that is key to rapid sensorimotor behaviors, and one that has important implications for autism given recent empirical and theoretical work implicating atypical prediction mechanisms and atypical reliance on priors in ASD.

The authors test this relationship in two different experiments, both of which show larger errors/biases in spatial updating for those with heightened autistic traits (as measured by AQ in neurotypical (NT) individuals).

Strengths:

The empirical results are convincing - effects are strong, sample sizes are sufficient, and the authors also rule out alternative explanations (ruling out differences in motor behavior or perceptual processing per se).

Weaknesses:

My main residual concern is that the paper should be more transparent about both (1) that this study does not include individuals with autism, and (2) acknowledging the limitations of the AQ.

On the first point, and I don't think this is intentional, there are several instances where the line between heightened autistic traits in the NT population and ASD is blurred or absent. For example, in the second sentence of the abstract, the authors state "Here, we examine the idea that sensory overload in ASD may be linked to issues with efference copy mechanisms". I would say this is not correct because the authors did not test individuals with ASD. I don't see a problem with using ASD to motivate and discuss this work, but it should be clear in key places that this was done using AQ in NT individuals.

For the second issue, the AQ measure itself has some problems. For example, reference 38 in the paper (a key AQ paper) also shows that the AQ is skewed more male than modern estimates of ASD, suggesting that the AQ may not fully capture the full spectrum of ASD symptomatology.

Of course, this does not mean that the AQ is not a useful measure (the present data clearly show that it captures something important about spatial updating during eye movements), but it should not be confused with ASD, and its limitations need to be acknowledged. My recommendation would be to do this in the title as well - e.g. note impaired visuomotor updating in individuals with "heightened autistic traits".

Suggestions for improvement:

- Figure 5 is really interesting. I think it should be highlighted a bit more, perhaps even with a model that uses the results of both tasks to predict AQ scores.

- Some discussion of the memory demands of the tasks will be helpful. The authors argue that memory is not a factor, but some support for this is needed.

- With 3 sessions for each experiment, the authors also have data to look at learning. Did people with high AQ get better over time, or did the observed errors/biases persist throughout the experiment?

---

## [Referee Report · Reviewer #2 (Public Review)]

Summary:

The idea that various clinical conditions may be associated, at least partially, with a disrupted corollary discharge mechanism has been present for long. In this paper, the authors draw a link between sensory overload, a characteristic of autism spectrum disorder, and a disturbance in the corollary discharge mechanism. The authors substantiate their hypothesis with strong evidence from both the motor and perceptual domains. As a result, they broaden the clinical relevance of the corollary discharge mechanism to encompass autism spectrum disorder.

Public comments:

The authors write:

"Imagine a scenario in which you're watching a video of a fast-moving car on a bumpy road. As the car hits a pothole, your eyes naturally make quick, involuntary saccades to keep the car in your visual field. Without a functional efference copy system, your brain would have difficulty accurately determining the current position of your eye in space, which in turn affects its ability to anticipate where the car should appear after each eye movement."

I appreciate the use of examples to clarify the concept of efference copy. However, I believe this example is more related to a gain-field mechanism, informing the system about the position of the eye with respect to the head, rather than an example of efference copy per-se.

Without an efference copy mechanism, the brain would have trouble to accurately determine where the eyes will be in space after an eye movement, and it will have trouble predicting the sensory consequences of the eye movement. But it can be argued that the gain-field mechanism would be sufficient to inform the brain about the current position of the eyes with respect the head.

The authors write:

"In the double-step paradigm, two consecutive saccades are made to briefly displayed targets 21,22. The first saccade occurs without visual references, relying on internal updating to determine the eye's position."

Maybe I am missed something, but in the double-step paradigm the first saccade can occur without the help of visual references if no visual feedback is present, that is, when saccades are performed in total darkness. Was this the case for this experiment? I could not find details about room conditions in the methods. Please provide further details.

In case saccades were not performed in total darkness, then the first saccade can be based on the remembered location of the first target presented, which can be derived from the retinotopic trace of the first stimuli, as well as contribution from the surroundings, that is: the remembered relative location of the first target with respect to the screen border along the horizontal meridian (i.e. allocentric cues)

A similar logic could be applied to the second saccade. If the second saccade were based only on the retinotopic trace, without updating, then it would go up and 45 deg to the right, based on the example shown in Figure 1. With appropriate updating, the second saccade would go straight up. However, if saccades were not performed in total darkness, then the location of the second target could also be derived from its relationship with the surroundings (for example, the remembered distance from screen borders, i.e. allocentric cues).

If saccades were not performed in total darkness, the results shown in Figures 2 and 3 could then be related to: (i) differences in motor updating between AQ score groups; (ii) differences in the use of allocentric cues between AQ score groups; (iii) a combination of (i) and (ii). I believe this is a point worth mentioning in the discussion."

The authors write:

"According to theories of saccadic suppression, an efference copy is necessary to predict the occurrence of a saccade."

I would also refer to alternative accounts, where saccadic suppression appears to arise as early as the retina, due to the interaction between the visual shift introduced by the eye movement, and the retinal signal associated with the probe used to measure saccadic suppression. This could potentially account for the scaling of saccadic suppression magnitude with saccade amplitude.

Idrees, S., Baumann, M.P., Franke, F., Münch, T.A. and Hafed, Z.M., 2020. Perceptual saccadic suppression starts in the retina. Nature communications, 11(1), p.1977.

---

## [Referee Report · Reviewer #3 (Public Review)]

Summary:

This work examined efference copy related to eye movements in healthy adults who have high autistic traits. Efference copies allow the brain to make predictions about sensory outcomes of self-generated actions, and thus serve important roles in motor planning and maintaining visual stability. Consequently, disrupted efference copies have been posited as a potential mechanism underlying motor and sensory symptoms in psychopathology such as Autism Spectrum Disorder (ASD), but so far very few studies have directly investigated this theory. Therefore, this study makes an important contribution as an attempt to fill in this knowledge gap. The authors conducted two eye-tracking experiments examining the accuracy of motor planning and visual perception following a saccade, and found that participants with high autistic traits exhibited worse task performance (i.e., less accurate second saccade and biased perception of object displacement), consistent with their hypothesis of less impact of efference copies on motor and visual updating. Moreover, the motor and visual biases are positively correlated, indicative of a common underlying mechanism. These findings are promising and can have important implications for clinical intervention, if they can be replicated in a clinical sample.

Strengths:

The authors utilized well-established and rigorously designed experiments and sound analytic methods. This enables easy translations between similar work in non-human primates and humans and readily points to potential candidates for underlying neural circuits that could be further examined in follow-up studies (e.g., superior colliculus, frontal eye fields, mediodorsal thalamus). The finding of no association between initial saccade accuracy and level of autistic trait in both experiments also serves as an important control analysis and increases one's confidence in the conclusion that the observed differences in task performance were indeed due to disrupted efference copies, not confounding factors such as basic visual/motor deficits or issues with working memory. The strong correlation between the observed motor and visual biases further strengthens the claim that the findings from both experiments may be explained by the same underlying mechanism - disrupted efference copies. Lastly, the authors also presented a thoughtful and detailed mechanistic theory of how efference copy impairment may lead to ASD symptomatology, which can serve as a nice framework for more research into the role of efference copies in ASD.

Weaknesses:

Although the paper has a lot of strengths, the main weakness of the paper is that a direct link with sensory/motor symptoms cannot be established. As the authors have discussed, the most likely symptoms resulting from disrupted efference copies would be sensory overload and motor inflexibility. The measure used to quantify the level of autistic traits, Autistic Quotient (AQ), does not capture any sensory or motor characteristics of the Autism spectrum. Therefore, it is unknown whether those scored high on AQ in this study experienced high, or even any, sensory or motor difficulties. In other words, more evidence is needed to demonstrate a direct link between disrupted efference copies and sensory/motor symptoms in ASD.

---

## [Author Response]

The following is the authors’ response to the original reviews.

**eLife assessment**
This important study tests the hypothesis that a high autism quotient in neurotypical adults is strongly associated with suboptimal motor planning and visual updating after eye movements, which in turn, is related to a disrupted efference copy mechanism. The implication is that such abnormal behavior would be exaggerated in those with ASD and may contribute to sensory overload - a key symptom in this condition. The evidence presented is convincing, with significant effects in both visual and motor domains, adequate sample sizes, and consideration of alternatives. However, the study would be strengthened with minor but necessary corrections to methods and statistics, as well as a moderation of claims regarding direct application to ASD in the absence of testing such patients.

**Public Reviews:**

**Reviewer #1 (Public Review):**
Summary:This study examines a hypothesized link between autism symptomatology and efference copy mechanisms. This is an important question for several reasons. Efference copy is both a critical brain mechanism that is key to rapid sensorimotor behaviors, and one that has important implications for autism given recent empirical and theoretical work implicating atypical prediction mechanisms and atypical reliance on priors in ASD.The authors test this relationship in two different experiments, both of which show larger errors/biases in spatial updating for those with heightened autistic traits (as measured by AQ in neurotypical (NT) individuals).Strengths:The empirical results are convincing - effects are strong, sample sizes are sufficient, and the authors also rule out alternative explanations (ruling out differences in motor behavior or perceptual processing per se).Weaknesses:My main concern is that the paper should be more transparent about both (1) that this study does not include individuals with autism, and (2) acknowledging the limitations of the AQ.On the first point, and I don't think this is intentional, there are several instances where the line between heightened autistic traits in the NT population and ASD is blurred or absent. For example, in the second sentence of the abstract, the authors state "Here, we examine the idea that sensory overload in ASD may be linked to issues with efference copy mechanisms". I would say this is not correct because the authors did not test individuals with ASD. I don't see a problem with using ASD to motivate and discuss this work, but it should be clear in key places that this was done using AQ in NT individuals.For the second issue, the AQ measure itself has some problems. For example, reference 38 in the paper (a key paper on AQ) also shows that those with high AQ skew more male than modern estimates of ASD, suggesting that the AQ may not fully capture the full spectrum of ASD symptomatology. Of course, this does not mean that the AQ is not a useful measure (the present data clearly show that it captures something important about spatial updating during eye movements), but it should not be confused with ASD, and its limitations need to be acknowledged. My recommendation would be to do this in the title as well - e.g. note impaired visuomotor updating in individuals with "heightened autistic traits".

We thank the reviewer for the kind words. We now specify more carefully that our sample of participants consists of neurotypical adults scored for autistic traits and none of them was diagnosed with autism before participating in our experiment. Regarding the Autistic Quotient Questionnaire (AQ) on page 5 of the Introduction we now write:

“The autistic traits of the whole population form a continuum, with ASD diagnosis usually situated on the high end 31-33. Moreover, autistic traits share a genetic and biological etiology with ASD 34. Thus, quantifying autistic-trait-related differences in healthy people can provide unique perspectives as well as a useful surrogate for understanding the symptoms of ASD 31,35.”

In the Discussion (page 9) we now write:

”It is essential to note that our participant pool lacked pre-existing diagnoses before engaging in the experiments and we must address limitations associated with the AQ questionnaire. The AQ questionnaire demonstrates adequate test-retest reliability 36, normal distribution of sum scores in the general population 50, and cross-cultural equivalence has been established in Dutch and Japanese samples 51-53. The AQ effectively categorizes individuals into low, average, and high degrees of autistic traits, demonstrating sensitivity for both group and individual assessments 54.

However, evolving research underscores many aspects that are not fully captured by the self-administered questionnaire: for example, gender differences in ASD trait manifestation 55. Autistic females may exhibit more socially typical interests, often overlooked by professionals 56. Camouflaging behaviors, employed by autistic women to blend in, pose challenges for accurate diagnosis 57. Late diagnoses are attributed to a lack of awareness, gendered traits, and outdated assessment tools 58. Moving forward, complementing AQ evaluations in the general population with other questionnaires, such as those assessing camouflaging abilities 59, or motor skills in everyday situation (MOSES-test 60) becomes crucial for a comprehensive understanding of autistic traits.”

Suggestions for improvement:- Figure 5 is really interesting. I think it should be highlighted a bit more, perhaps even with a model that uses the results of both tasks to predict AQ scores.

We thank the reviewer for the suggestion. However, the sample size is relatively small for building a robust and generalizable model to predict AQ scores. Statistical models built on small datasets can be prone to overfitting, meaning that they might not accurately predict the AQ for new individuals.

- Some discussion of the memory demands of the tasks will be helpful. The authors argue that memory is not a factor, but some support for this is needed.

The reviewer raises an important point regarding the potential for memory demands to influence our results. We have now also investigated the accuracy of the second saccade separately for the x and y dimension. As also shown in figure 3 panel A, a motor bias was observed only in one dimension (x), weaking the argument of memory which would imply a bias in both directions (participants remembering the position of the target relative to both screen borders for example). We performed a t-test between our subsample of participants and indeed we found a difference in saccade accuracy for the x dimension (p = 0.03) but not in the y dimension (p = 0.88).

We now add these analyses in Discussion on page 8.

- With 3 sessions for each experiment, the authors also have data to look at learning. Did people with high AQ get better over time, or did the observed errors/biases persist throughout the experiment?

We thank the reviewer for pointing this out. On page 7 (Results) we now write:

” Understanding how these biases might change over time could provide further insights into this mechanism. Specifically, we investigated whether participants exhibited any learning effects throughout the experiments. For data of Experiment 1 – motor updating – we divided our data into 10 separate bins of 30 trials each. We conducted a repeated measure ANOVA with the within-subject factor “number of sessions” (two main sessions of 5 bins each, ~150 trials) and the between-subject factor “group” (lower vs upper quartile of the AQ distribution). We found no main effect of “number of sessions” (F(1,7) = 0.25, p = 0.66), a main effect of “group” (F(1,7) = 2.52, p = 0.015), and no interaction between the two subsample of participants and the sessions tested (F(1,7) = 0.51, p = 0.49). Data of Experiment 2 – visual updating– were separated into 3 sessions. For each session we extracted the PSE and we conducted a repeated measure ANOVA with within subject factor “sessions” and between subject factor “groups” (lower vs upper quartile of the AQ distribution). Also here we found no main effect of sessions (F(1,13) = 0.86, p = 0.39), a main effect of group (F(1,14) = 11.85, p = 0.004), and no interaction between the two subsample of participants and the sessions tested (F(1,13) = 0.20, p = 0.73). In conclusion, the current study found no evidence of learning effects across the experimental sessions. However, a significant main effect of group was observed in both Experiment 1 (motor updating) and Experiment 2 (visual updating). Participants in the group with higher autistic traits performed systematically differently on the task, regardless of the number of sessions completed compared to those in the group with lower autistic traits.”

**Reviewer #2 (Public Review):**
Summary:The idea that various clinical conditions may be associated, at least partially, with a disrupted corollary discharge mechanism has been present for a long time.In this paper, the authors draw a link between sensory overload, a characteristic of autism spectrum disorder, and a disturbance in the corollary discharge mechanism. The authors substantiate their hypothesis with strong evidence from both the motor and perceptual domains. As a result, they broaden the clinical relevance of the corollary discharge mechanism to encompass autism spectrum disorder.The authors write:"Imagine a scenario in which you're watching a video of a fast-moving car on a bumpy road. As the car hits a pothole, your eyes naturally make quick, involuntary saccades to keep the car in your visual field. Without a functional efference copy system, your brain would have difficulty accurately determining the current position of your eye in space, which in turn affects its ability to anticipate where the car should appear after each eye movement."I appreciate the use of examples to clarify the concept of efference copy. However, I believe this example is more related to a gain-field mechanism, informing the system about the position of the eye with respect to the head, rather than an example of efference copy per se.Without an efference copy mechanism, the brain would have trouble accurately determining where the eyes will be in space after an eye movement, and it will have trouble predicting the sensory consequences of the eye movement. However it can be argued that the gain-field mechanism would be sufficient to inform the brain about the current position of the eyes with respect to the head.

We now used a different example. And on page 3 of Introduction, we now write:

“During a tennis game, rapid oculomotor saccades are employed to track the high-velocity ball across the visual display. In the absence of a functional efference copy mechanism, the brain would encounter difficulty in anticipating the precise retinal location of the ball following each saccade. This could result in a transient period of visual disruption as the visual system adjusts to the new eye position. The efference copy, by predicting the forthcoming sensory consequences of the saccade, would bridge this gap and facilitate the maintenance of a continuous and accurate representation of the ball's trajectory.”

The authors write:"In the double-step paradigm, two consecutive saccades are made to briefly displayed targets 21, 22. The first saccade occurs without visual references, relying on internal updating to determine the eye's position."Maybe I have missed something, but in the double-step paradigm the first saccade can occur without the help of visual references if no visual feedback is present, that is, when saccades are performed in total darkness. Was this the case for this experiment? I could not find details about room conditions in the methods. Please provide further details.In case saccades were not performed in total darkness, then the first saccade can be based on the remembered location of the first target presented, which can be derived from the retinotopic trace of the first stimuli, as well as the contribution from the surroundings, that is: the remembered relative location of the first target with respect to the screen border along the horizontal meridian (i.e. allocentric cues).A similar logic could be applied to the second saccade. If the second saccade were based only on the retinotopic trace, without updating, then it would go up and 45 deg to the right, based on the example shown in Figure 1. With appropriate updating, the second saccade would go straight up. However, if saccades were not performed in total darkness, then the location of the second target could also be derived from its relationship with the surroundings (for example, the remembered distance from screen borders, i.e. allocentric cues).If saccades were not performed in total darkness, the results shown in Figures 2 and 3 could then be related to (i) differences in motor updating between AQ score groups; (ii) differences in the use of allocentric cues between AQ score groups; (iii) a combination of (i) and (ii). I believe this is a point worth mentioning in the discussion."

Thank you for raising the important issue of visual references in the double-step saccade task. Participants performed saccades in a dimly lit room where visual references, i.e. the screen borders, were barely visible. At the time we collected the data a laboratory that allowed performing experiments in complete darkness was not at our disposal. We acknowledge the possibility that participants could have memorized the target locations relative to the screen borders. The bias of high AQ participants could then be attributed to differences in either encoding, memorization or decoding of the target location relative to the screen borders. However, the potentially abnormal use of visual references must reflect an altered remapping process since we did not find differences in saccade landing in the vertical dimension. A t-test between our group of participants revealed a difference in saccade accuracy for the x dimension (p = 0.03) but not in the y dimension (p = 0.88). We thus agree that in addition to an altered efference copy signal in high AQ participants, altered use of visual references might also affect their saccadic remapping.

In Discussion we now write: “Our findings suggest that a general memory deficit is unlikely to fully explain the observed bias in high-AQ participants' second saccades. As highlighted in Figure 3A, the bias was specific to the horizontal dimension, weakening the argument for a global memory issue affecting both vertical and horizontal encoding of target location. However, it's important to acknowledge that even under non-darkness conditions, participants might rely on a combination of internal updating based on the initial target location and visual cues from the environment, such as screen borders. This potential use of visual references could contribute to the observed bias in the high-AQ group. If high-AQ participants differed in their reliance on visual cues compared to the low-AQ group, it could explain the specific pattern of altered remapping observed in the horizontal dimension. This possibility aligns with our argument for an abnormal remapping process underlying the results. While altered efference copy signals remain a strong candidate, the potential influence of visual cues on remapping in this population warrants further investigation. Future studies could incorporate a darkness condition to isolate the effects of internal updating on the first saccade, and systematically manipulate the availability of visual cues throughout the task. This would allow for a more nuanced understanding of how internal updating and visual reference use interact in the double-step paradigm, particularly for individuals with varying AQ scores “.

The authors write:According to theories of saccadic suppression, an efference copy is necessary to predict the occurrence of a saccade."I would also refer to alternative accounts, where saccadic suppression appears to arise as early as the retina, due to the interaction between the visual shift introduced by the eye movement, and the retinal signal associated with the probe used to measure saccadic suppression. This could potentially account for the scaling of saccadic suppression magnitude with saccade amplitude.Idrees, S., Baumann, M.P., Franke, F., Münch, T.A. and Hafed, Z.M., 2020. Perceptual saccadic suppression starts in the retina. Nature communications, 11(1), p.1977.

We thank the reviewer. Now on page 4 of Introduction we write:

“Some theories consider saccadic omission and saccadic suppression as resulting from an active mechanism. In this view an efference copy would signal the occurrence of a saccade, yielding a transient decrease in visual sensitivity20-22. Others however have pointed out the possibility that a purely passive mechanism suffices to induce saccadic omission23. A recent study has found evidence for saccadic suppression already in the retina. Idrees et al.24 demonstrated that retinal ganglion cells in isolated retinae of mice and pigs respond to saccade-like displacements, leading to the suppression of responses to additional flashed visual stimuli through visually triggered retinal-circuit mechanisms. Importantly, their findings suggest that perisaccadic modulations of contrast sensitivity may have a purely visual origin, challenging the need for an efference copy in the early stages of saccadic suppression. However, the suppression they measured lasted much longer than time-courses observed in behavioral data. An efference copy signal could thus be necessary to release perception from suppression.”

**Reviewer #3 (Public Review):**
Summary:This work examined efference copy related to eye movements in healthy adults who have high autistic traits. Efference copies allow the brain to make predictions about sensory outcomes of self-generated actions, and thus serve important roles in motor planning and maintaining visual stability. Consequently, disrupted efference copies have been posited as a potential mechanism underlying motor and sensory symptoms in psychopathology such as Autism Spectrum Disorder (ASD), but so far very few studies have directly investigated this theory. Therefore, this study makes an important contribution as an attempt to fill in this knowledge gap. The authors conducted two eye-tracking experiments examining the accuracy of motor planning and visual perception following a saccade and found that participants with high autistic traits exhibited worse task performance (i.e., less accurate second saccade and biased perception of object displacement), consistent with their hypothesis of less impact of efference copies on motor and visual updating. Moreover, the motor and visual biases are positively correlated, indicative of a common underlying mechanism. These findings are promising and can have important implications for clinical intervention if they can be replicated in a clinical sample.Strengths:The authors utilized well-established and rigorously designed experiments and sound analytic methods. This enables easy translations between similar work in non-human primates and humans and readily points to potential candidates for underlying neural circuits that could be further examined in follow-up studies (e.g., superior colliculus, frontal eye fields, mediodorsal thalamus). The finding of no association between initial saccade accuracy and level of autistic trait in both experiments also serves as an important control analysis and increases one's confidence in the conclusion that the observed differences in task performance were indeed due to disrupted efference copies, not confounding factors such as basic visual/motor deficits or issues with working memory. The strong correlation between the observed motor and visual biases further strengthens the claim that the findings from both experiments may be explained by the same underlying mechanism - disrupted efference copies. Lastly, the authors also presented a thoughtful and detailed mechanistic theory of how efference copy impairment may lead to ASD symptomatology, which can serve as a nice framework for more research into the role of efference copies in ASD.Weaknesses:Although the paper has a lot of strengths, the main weakness of the paper is that a direct link with ASD symptoms (i.e., sensory overload and motor inflexibility as the authors suggested) cannot be established. First of all, the participants are all healthy adults who do not meet the clinical criteria for an ASD diagnosis. Although they could be considered a part of the broader autism phenotype, the results cannot be easily generalized to the clinical population without further research. Secondly, the measure used to quantify the level of autistic traits, Autistic Quotient (AQ), does not actually capture any sensory or motor symptoms of ASD. Therefore, it is unknown whether those who scored high on AQ in this study experienced high, or even any, sensory or motor difficulties. In other words, more evidence is needed to demonstrate a direct link between disrupted efference copies and sensory/motor symptoms in ASD.

This is a valid point, and we thank the reviewer for raising it up. Moving forward, complementing AQ evaluations in the general population with other questionnaires, such as those assessing camouflaging abilities (Hull, L., Mandy, W., Lai, MC., et al., 2019), or motor skills in everyday situation (MOSES-test, Hillus J, Moseley R, Roepke S, Mohr B. 2019 ) becomes crucial for a comprehensive understanding of autistic traits.”

We now address this point in Discussion page 9.

**Recommendations for the authors:**

**Reviewer #1 (Recommendations For The Authors):**
Minor comments- The pothole example in the introduction was really hard to follow. I wonder if there is a better example.

We now used a different example. And on page 3 of Introduction, we now write:

“During a tennis game, rapid oculomotor saccades are employed to track the high-velocity ball across the visual display. In the absence of a functional efference copy mechanism, the brain would encounter difficulty in anticipating the precise retinal location of the ball following each saccade. This could result in a transient period of visual disruption as the visual system adjusts to the new eye position. The efference copy, by predicting the forthcoming sensory consequences of the saccade, would bridge this gap and facilitate the maintenance of a continuous and accurate representation of the ball's trajectory.”

- This is really minor; I would say that saccades are not the most frequent movement that humans perform. Some of the balance-related adjustments and even heartbeats are faster. Maybe just add "voluntary".

We thank the reviewer for the suggestion, now added.

- "Severe consequences" on page 4 is a bit strong. If that were true, there would be pretty severe impairments in eye movement behavior in ASD, which I don't think is the case.

We agree with the reviewer. We now eliminated the term “severe”.

- The results section would read better if each experiment had a short paragraph reiterating its overall goal and the specific approach each experiment took to achieve that goal.

Now on page 5, for the first experiment, we write:

”We investigated the influence of autistic traits on visual updating during saccadic eye movements using a classic double-step saccade task. This task relies on participants making two consecutive saccades to briefly presented targets. The accuracy of the second saccade serves as an indirect measure of how effectively the participant's brain integrated the execution of the first saccade into their internal representation of visual space. Participants were divided into quartiles based on the severity of their autistic traits, as assessed by the Autistic quotient questionnaire (cite). We hypothesized that individuals with higher autistic traits would exhibit greater difficulty in visual updating compared to those with lower autistic traits. This would be reflected in reduced accuracy of their second saccades in the double-step task. Figure 2C illustrates examples from participants at the extremes of the autistic trait distribution (Autistic quotient = 3, in orange and Autistic quotient = 31, in magenta). As shown, both participants were instructed to make saccades to the locations indicated by two brief target appearances (T1 and T2), as quickly and accurately as possible, following the order of presentation. However, successful execution of the second saccade requires accurate internal compensation for the first saccade, without any visual references or feedback available during the saccade itself.”

On page 6, for experiment 2, we write:

”With a trans-saccadic localization task, we explored how autistic traits affect the integration of eye movements into visual perception. Participants were presented with stimuli before and after a single saccade, creating an illusion of apparent motion. We measured the perceived direction of this displacement, which is influenced by how well the participant's brain accounts for the saccadic eye movement. We predicted that individuals with higher autistic traits would show a stronger bias in the perceived displacement direction, suggesting a less accurate integration of the eye movement into their visual perception.”

- On page 6, the text about "vertical displacement" is confusing. The spatial displacements in this experiment were horizontal?

Yes, they were. The spatial displacement is horizontal, but the perceived trajectory (due to the saccade) is vertical. We now changed “vertical displacement” to “vertical trajectory”.

- Page 6, grammatical problems in "while we report a slightly slant of the dots trajectory".

Thank you. Now fixed.

- It would be helpful to discuss the apparent motion part of Experiment 2 in the main text. This important part is not made clear.

We now in Introduction, page 4, write:

“In this paradigm, one stimulus is shown before and another after saccade execution. Together these two stimuli produce the perception of “apparent motion”. If stimuli are placed such that the apparent motion path is orthogonal to the saccade path, then the orientation of the apparent motion path indicates how the saccade vector is integrated into vision. The apparent motion trajectory can only appear vertical if the movement of the eyes is perfectly accounted for, that is the retinotopic displacement is largely compensated, ensuring spatial stability. However, small biases of motion direction – implying under- (or over-) compensation of the eye movement – can indicate relative failures in this stabilization process. In a seminal study, Szinte and Cavanagh 27 found a slight over-compensation of the saccade vector leading to apparent motion slightly tilted against the direction of the saccade. More importantly, when efference copies are not available, i.e. localization occurring at the time of a second saccade in a double step task, a strong saccade under-compensation occurs 28.

This phenomenon cannot be explained by perisaccadic mislocalization of flashed visual stimuli 29,30, but the two phenomena may be related in that they may both depend upon efference copy information.”

- Figure 1 could be improved. For example, the text talks about the motor plan, but this is not clearly shown in the figure.

We now added the motor plan into the model. Thank you.

- Figure 2A, the scale is off (the pictures make it look like the horizontal movement was longer than the vertical).

Now fixed.

- Figure 4, it would be helpful if the task was also described in the figure.

We thank the reviewer for the comment. We now tried to modify the figure by also adding the perceptual judgment task.

- Figure 5A, the y-axis shows p(correct), but that is not what the y-axis shows (the legend makes the same mistake).

We apologize, it’s the proportion of time participants reported the second dot to be more to the right compared to the first one. We now changed the figure and the text accordingly.

- A recent study on motion and eye movement prediction in ASD is very relevant to the work presented here.: Park et al. (2021). Atypical visual motion-prediction abilities in autism spectrum disorder. Clinical Psychological Science, 9(5), 944-960.

Indeed. We now refer to the cited study in Discussion, on page 9.

**Reviewer #2 (Recommendations For The Authors):**
Statistics and plotting.I believe some of the reported statistics are not clear. For example, the authors write:"Saccade landing positions of participants in the lower quartile (mean degree {plus minus} SEM: 10.17{plus minus} 0.50) did not deviate significantly from those in the upper quartile (mean degree {plus minus} SEM: 9.65 {plus minus} 0.77). This result was also confirmed by a paired sample t-test (t(7) = 0.66; p = 0.66, BF10 = 0.40)"Maybe I am missing something, but why use a paired-sample t-test when the upper and lower quartiles constitute different groups of participants? Shouldn't a two-sample t-test be used in this case?

We apologize for the confusion. It is indeed a two-sample t-test.

Along the same lines, I do not understand the link between the number of degrees of freedom reported in the t-test (7) and the number of participants reported in the study (41).This is also evident when looking at the scatterplot in Figure 3C. How many participants formed the averages and standard errors reported in Figures 3B and 3D? Please clarify.I have the same comment(s) also for the visual updating task (and related figures), where 13 degrees of freedom are reported in the t-tests. Please clarify.

We thank the reviewer for pointing this out. The number of participants reported in the scatter plots were indeed 42. However, we opted to compare the averages only in the lower and upper quartile of the AQ distribution to avoid dealing with a median split (which would imply a skewed distribution). Of our sample of participants in Exp1, 8 fell into the lower quartile of the AQ distribution and 8 in the upper quartile (14 deg of freedom); from Exp 2, 8 participants fell in the lower and 7 in the upper (13 deg of freedom).

We now fixed the values accordingly.

**Reviewer #3 (Recommendations For The Authors):**
(1) The language can be a bit misleading (especially the title and abstract) as it wasn't always clear that the participants don't actually have clinical ASD. I'd suggest avoiding using words like "symptom" as that would indicate clinical severity, and using words like "traits/characteristics" instead for more precise language.

We apologize for the misleading terminology used. Now fixed.

(2) In the Intro: "...perfect compensation results in a vertical trajectory, while small biases indicate stabilization issues23-25." This is a bit confusing without knowing the details of the paradigm. Consider clarifying or at least referring to Figure 4.

Thank you.

(3) In the Results: "This result was also confirmed by a paired sample t-test (t(7) = 0.66;..." This is confusing as a two-sample t-test is the appropriate test here. Also, the degree of freedom seems very low - could the authors clarify how many participants are in each subgroup (i.e., low vs. high AQ quartile), for both experiments?

Of our sample of participants in Exp1 8 fell into the lower quartile of the AQ distribution and 8 in the upper quartile (14 deg of freedom); from Exp 2, 8 participants fell in the lower and 7 in the upper (13 deg of freedom).

(4) In the Methods: Experiment 2: "The first dot could appear randomly above or below gaze level at a fixed horizontal location, halfway between the two fixations (x = 0, y = -5{degree sign} or +5{degree sign} depending on the trial). The second dot was then shown orthogonal to the first one at a variable horizontal location (x = 5{degree sign} {plus minus} 2.5{degree sign})." This would mean that the position of the 2nd dot relative to the 1st one would be 2.5{degree sign}- 7.5{degree sign}, but the task description in Results and Figure 5A would suggest the horizontal location of the second dot is x = 0{degree sign} {plus minus} 2.5{degree sign}. Which one is correct?

The second option is the correct one. We now fixed the typo in the Methods part.

(5) There is another study that examined oculomotor efference copies in children with ASD using a similar trans-saccadic perception task (Yao et al., 2021, Journal of Vision). In that study, they found a correlation between task performance and an ASD motor symptom (repetitive behavior). This seems quite relevant to the authors' hypothesis and discussion.

We thank the reviewer for the suggestion. We now added the mentioned paper in the discussion.

(6) Please proofread the entire paper carefully as there were multiple grammatical and spelling errors.

Thank you.